# Covalent organic frameworks with high quantum efficiency in sacrificial photocatalytic hydrogen evolution

Chunzhi Li[1,2], Jiali Liu[1,2], He Li [1✉], Kaifeng Wu [3], Junhui Wang [3✉] & Qihua Yang [1✉]

Organic semiconductors offer a tunable platform for photocatalysis, yet the more difficult exciton dissociation, compared to that in inorganic semiconductors, lowers their photo-catalytic activities. In this work, we report that the charge carrier lifetime is dramatically prolonged by incorporating a suitable donor-acceptor (β-ketene-cyano) pair into a covalent organic framework nanosheet. These nanosheets show an apparent quantum efficiency up to 82.6% at 450 nm using platinum as co-catalyst for photocatalytic $H_2$ evolution. Charge carrier kinetic analysis and femtosecond transient absorption spectroscopy characterizations verify that these modified covalent organic framework nanosheets have intrinsically lower exciton binding energies and longer-lived charge carriers than the corresponding nanosheets without the donor-acceptor unit. This work provides a model for gaining insight into the nature of short-lived active species in polymeric organic photocatalysts.

[1] State Key Laboratory of Catalysis, Dalian Institute of Chemical Physics, Chinese Academy of Sciences, 457 Zhongshan Road, Dalian 116023, China. [2] University of Chinese Academy of Sciences, Beijing 100049, China. [3] State Key Laboratory of Molecular Reaction Dynamics, Dalian Institute of Chemical Physics, Chinese Academy of Sciences, 457 Zhongshan Road, Dalian 116023, China. ✉email: lihe@dicp.ac.cn; wjh@dicp.ac.cn; yangqh@dicp.ac.cn

Photocatalytic water splitting for $H_2$ production is a sustainable way to convert solar energy into clean chemical energy, which can help to solve the current energy crisis and environmental issues. Since the 1970s, scientists all over the world have been making endless attempts to get this "holy grail"[1–7]. Very promisingly, Takata et al. reported that the apparent quantum efficiency (AQE) of almost unity was achieved over $SrTiO_3$ for overall water splitting irradiated with the light in the ultraviolet region by selecting a suitable co-catalyst and fully promoting the spatial separation of charge carriers among different crystal facets[8]. However, given that the visible light occupies 42–45% of the solar spectrum[9–11], the development of visible-light responsive photocatalysts is necessary for the efficient use of solar energy. Up to date, the inorganic semiconductors (ISs) dominate in the photocatalytic water splitting, but the visible-light responsive ISs with suitable band structure for water splitting is still very limited that is related to the difficulties in the band structure engineering.

Covalent organic frameworks (COFs) as a kind of organic polymers have gradually shined in the fields of gas separation[12–14], energy storage[15–18], sensors[19–22], and catalysis[23–30]. Especially, two-dimensional (2D) COFs with extended π-π conjugation structures have demonstrated great application potential in photocatalysis. One of their unique advantages is that the band structure of 2D COFs can be fine-tuned at the molecular level by incorporating of different organic building blocks. It is worth mentioning that although 2D COFs favor some essential photocatalysis steps such as light trapping, charge separation and charge carrier migration; most of 2D COFs only exhibit moderate photocatalytic activities, especially in comparison with ISs, which is possibly related to the high exciton binding energy and fast charge recombination[31]. It has been reported that the excitons dissociation ability of 2D COFs could be improved by incorporating donor–acceptor (D–A) structure, enhancing the network polarity and reinforcing the conjugation structure[32–34]. For this reason, triazine- and halogen-based COFs[35,36], and sp2 carbon-conjugated COFs[37,38] have been synthesized and they show enhanced activities in hydrogen evolution reaction (HER).

Cyano moiety (CYANO) as an electron-withdrawing group, has been wildly introduced in classical non-fullerene acceptors, such as ITIC and Y6[39,40]. The photovoltaic devices, such as ITO/PEDOT:PSS/PM6:Y6/PDINO/Al, exhibit efficient device performance with power conversion efficiencies up to 15.7%[41], largely because of choosing a suitable D–A structure to promote the separation of charges which is also a critical step in photocatalysis. The cyano moiety appears in some recently reported materials for photocatalytic HER, such as sp2 carbon-conjugated COFs based on Knoevenagel condensation reaction, covalent triazine frameworks based on benzonitrile trimerization reaction and cyano-containing conjugated polymer[42–44]. Still, none of these COFs show high AQE or photocatalytic HER activity, which is possibly related to the surface hydrophobicity and inefficient D–A pair.

Recently, Coopper et al. reported that conjugated polymers with sulfone building blocks are also very promising in photocatalytic HER due to their hydrophilicity and excellent charge separation properties[45–47]. It is also demonstrated that COFs nanosheets/nanolayers are more active in photocatalysis than bulk COFs due to the reduced charge carrier recombination in the short diffusion distance and also more exposing reaction surface[48,49]. More importantly, the understanding of exciton and charge carrier dynamics, which is a key step in photocatalysis[50–52], plays an important role in enhancing the photocatalytic activity of COFs.

In this work, we report the synthesis of a cyano-containing COF (CYANO-COF) with ketene-cyano (D–A) pair via a Schiff-base condensation reaction of 1,3,5-triformylphloroglucinol (Tp) with

4,4'-diamino-[1,1'-biphenyl]-3,3'-dicarbonitrile (BD-CYANO). CYANO-CON (COFs nanosheet obtained by ball milling of CYANO-COF) afforded a high AQE up to 82.6% at 450 nm, a record-breaking AQE for hydrogen evolution for the COF-based photocatalysts as far as we know. The whole picture of exciton dissociation and charge recombination is elucidated with temperature-dependent photoluminescence (PL) and femtosecond transient absorption (fs-TA) measurements.

## Results

**Designed synthesis and characterizations**. CYANO-COF was synthesized via a Schiff-base condensation reaction of 1,3,5-tri-formylphloroglucinol (Tp) with 4,4'-diamino-[1,1'-biphenyl]-3,3'-dicarbonitrile (BD-CYANO) in the presence of 6 M aqueous acetic acid (Fig. 1a). The formation of β-ketoenamine linkage via a keto-enol tautomerization could increase the chemical stability of CYANO-COF and the ketene can also serve as an electron donor. The computational study of charge distribution for CYANO-COF demonstrated that cyano and ketene serve as electron acceptor and donor, respectively (Fig. 1b). A control sample, BD-COF with similar linkage and topology structure to CYANO-COF but without cyano as acceptor, was synthesized according to a reported method using Tp and benzidine (BD) as monomers[53].

The FT-IR spectrum of CYANO-COF displayed vibration peaks at 1618 and 1579 $cm^{-1}$, respectively, assigned to the C=O and C=C stretching vibrations together with aromatic ring skeleton vibrations at 1494 and 1443 $cm^{-1}$, showing the formation of β-ketoenamine linkage (Fig. 1c)[53,54]. Notably, the typical C≡N stretching vibration appeared at 2200 $cm^{-1}$, indicating that cyano groups can endure the synthesis condition without decomposition. In addition, the $^{13}C$ CP-TOSS NMR spectrum of CYANO-COF provided strong supportive structural information with apparent chemical shift for -C=O at 184 ppm, -NH-C=C at 149 and 104 ppm, C≡N at 114 ppm and aromatic rings in the range of 150–95 ppm (Fig. 1d). All these data provided adequate chemical composition evidence for the successful preparation of CYANO-COF with cyano groups. The FT-IR spectrum of BD-COF was congruent with a previous report[53], showing the presence of vibrations associated with β-ketoenamine structure (Supplementary Fig. 1). CYANO-COF and BD-COF with decomposition temperatures beyond 350 °C in air flow displayed high thermal stability evaluated by thermogravimetric analysis (Supplementary Fig. 2).

The crystalline nature of CYANO-COF was characterized by the powder X-ray diffraction (PXRD) technique. The PXRD pattern of CYANO-COF exhibited a predominant peak at 3.60°, corresponding to the reflection of (100) plane, with other weak peaks at 6.25°, 7.18° and a broad peak at 26.5°, which can be assigned to the (110), (200) and (001) plane, respectively (Fig. 1e). Furthermore, Pawley refinement confirmed that the diffraction patterns of CYANO-COF were consistent with a hexagonal lattice with P6/M space group ($a = b = 28.78$ Å, $c = 3.60$ Å; $\alpha = \beta = 90°$, $\gamma = 120°$; Rp = 3.99%, Rwp = 4.91%) similar to an eclipsed model (Supplementary Table 1 and Fig. 1e). A poor correlation with crystallographic structures of CYANO-COF was obtained with the staggered AB model (Supplementary Table 2 and Supplementary Fig. 3), further confirming the eclipsed AA stacking model of CYANO-COF. Analogously, the PXRD patterns of BD-COF were consistent with a previous report[53], showing the eclipsed AA stacking model (Supplementary Fig. 4). In comparison with CYANO-COF, the (001) diffraction of BD-COF shifted to a higher 2-theta angle, indicating that the interlayer space of BD-COF was smaller than that of CYANO-COF, likely due to the increased charge repulsion between

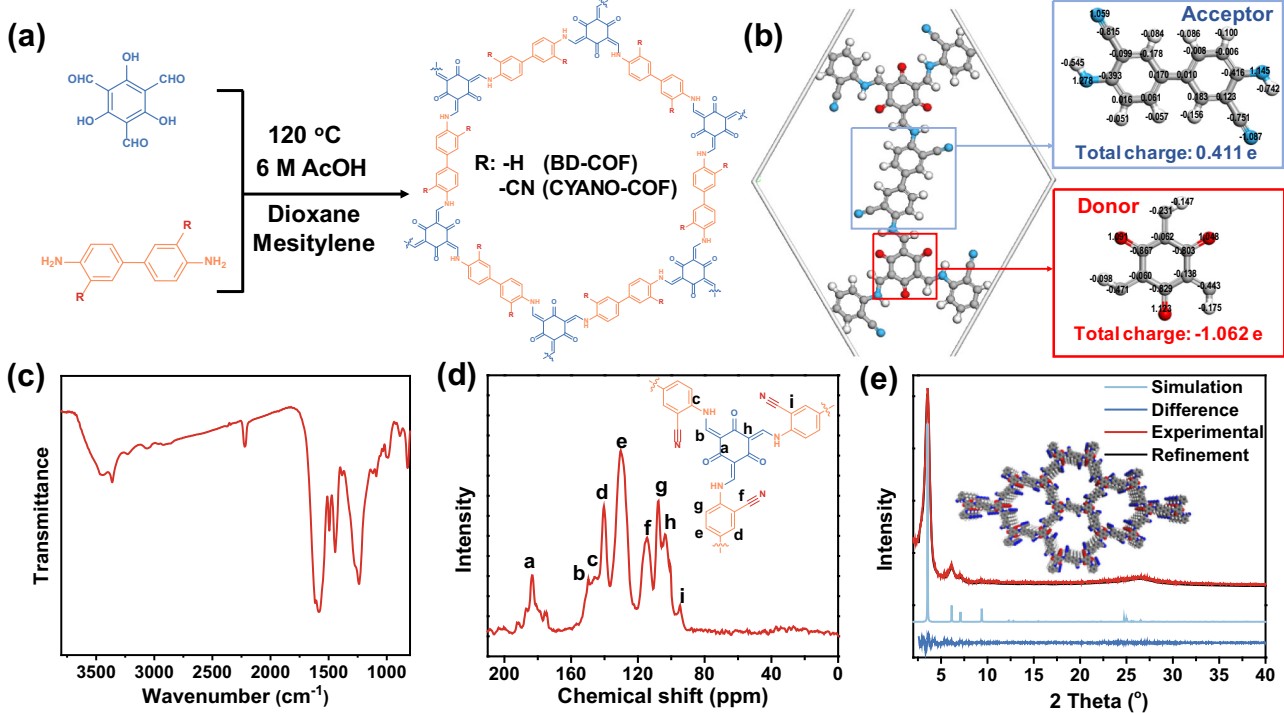

**Fig. 1 Chemical structure, charge distribution, and characterizations of CYANO-COF. a** Synthesis of CYANO-COF and BD-COF. **b** Charge distribution in CYANO-COF structure. **c** FT-IR spectrum of CYANO-COF. **d** Solid-state $^{13}$C CP-TOSS NMR spectrum of CYANO-COF. **e** Experimental, simulated, refined PXRD patterns and the refinement differences of CYANO-COF.

interlayer with the existence of a strong polar cyano group[55]. All organic semiconductors displayed quite different diffraction peaks with monomers, demonstrating the successful formation of the corresponding polymers without residual monomers (Supplementary Fig. 4).

The Brunauer–Emmett–Teller (BET) surface area of CYANO-COF measured by nitrogen sorption isotherms at 77 K was 559 m$^2$ g$^{-1}$ (Supplementary Fig. 5), lower than the predicted value (2667 m$^2$ g$^{-1}$) with the AA stacking model (Supplementary Fig. 6). Different experiment and theoretical values are generally observed for COFs possibly due to the structure distortion. The pore size of CYANO-COF is distributed from 1.0 to 2.5 nm calculated by the nonlocal density functional theory method (Supplementary Table 3 and Supplementary Fig. 5), which agreed well with a predicted AA stacking pore size (Supplementary Fig. 6). The BD-COF has a BET surface area of 519 m$^2$ g$^{-1}$ with a pore size distribution from 1.0 to 2.5 nm (Supplementary Table 3 and Supplementary Fig. 5).

Previous studies show that the layer thickness greatly affects the charge separation efficiency of COFs[48,49]. Therefore, we tried to prepare thin layered COFs by sonication method. The scanning electron microscopy images of both CYANO-COF and BD-COF depicted a rod-like morphology (Supplementary Fig. 7). The rod-like morphology of CYANO-COF remained unchanged even with sonication for 24 h, implying the high mechanical stability of CYANO-COF (Supplementary Fig. 7). Impressively, CYANO-CON and BD-CON were successfully obtained by ball milling of CYANO-COF and BD-COF with the assistance of sonication. The transmission electron microscopy (TEM) images of CYANO-CON and BD-CON showed almost identical nanosheet morphology with lateral sizes close to 500 nm (Supplementary Fig. 8). The periodic framework structure of CYANO-CON was visualized by high-resolution transmission electron microscopy (HRTEM). The ordered arrangement of mesopores could be clearly observed in the HRTEM image of

CYANO-CON (Fig. 2a). The Fourier-filtered image of the enlarged red square area showed that the interplanar spacing of (100) lattice plane was 2.1 nm, consistent with the pore size by N$_2$ sorption isotherm and simulated eclipsed model (Fig. 2b and Supplementary Fig. 6).

The atomic force microscopy (AFM) images of both CONs drop-coated onto mica from ethanol suspensions also displayed irregular nanosheet topography with thickness ranging from 4 to 5 nm, corresponding to the existence of only ~12–15 COF layers (Fig. 2c). The particle size of the nanosheet varied in the range of 200 nm to 3.5 μm by measuring 200 particles with AFM characterization on a large scale (30 μm, excluding a few elongated nanosheets, Supplementary Fig. 10). The PXRD patterns and pore size distributions of CYANO-CON and BD-CON after ball milling were identical to pristine COFs, indicating that COFs can endure the high-energy ball milling process because of the high thermostability (Supplementary Fig. 4). The BET surface area of CONs decreased as compared with the pristine COFs due to the exfoliated ultra-thin nanosheet effect[56]. Both CONs can be well dispersed in water to form colloid solutions as verified by the conspicuous Tyndall effect. These colloid solutions could remain stable even over 4 months (Fig. 2e and Supplementary Fig. 9). The dynamic light scattering measurement showed the dominant colloid size distribution at ~530 nm for both CONs, which coincided with corresponding TEM and AFM results. All these structural pieces of evidence proved that the 2D CONs were successfully obtained by mechanical exfoliation of bulk COFs.

The chemical stability of CYANO-COF under harsh conditions, especially prolonged light irradiation, is the prerequisite for photocatalysis application. Interestingly, CYANO-COF could withstand different harsh conditions, such as a 3-day immersion in THF, DMSO, DMF, 3 M aqueous HCl, and 3 M NaOH, as evidenced by the almost identical PXRD patterns and FT-IR spectra before and after treatments (Supplementary Fig. 11). Even

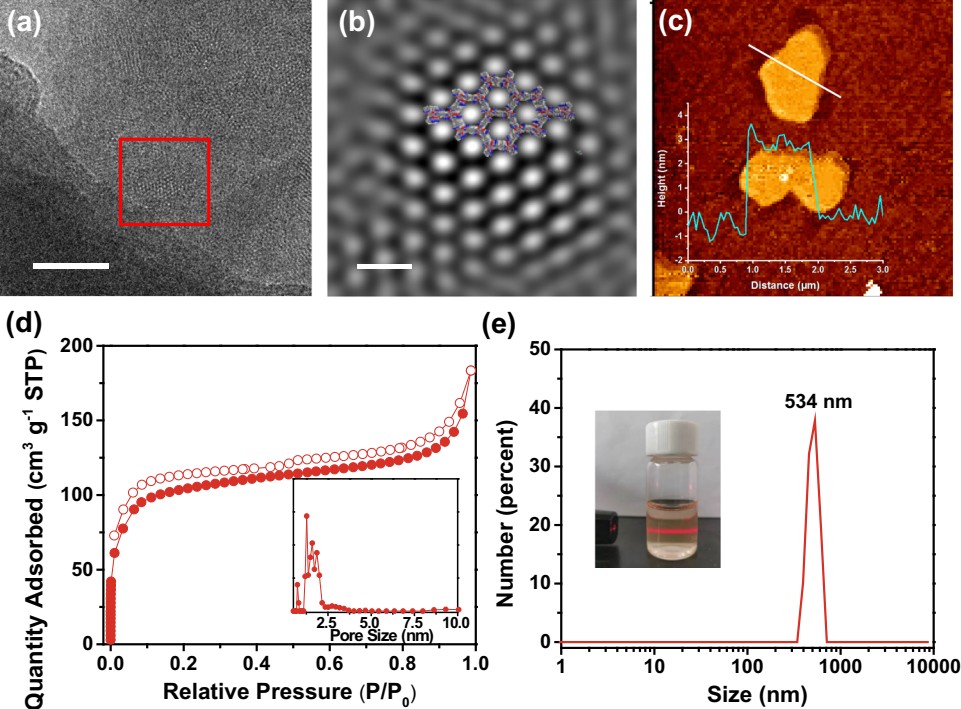

**Fig. 2 Characterizations of CYANO-CON (obtained by ball milling of CYANO-COF). a** HRTEM image (scale bar, 50 nm), and **b** Fourier-filtered image of a selected red square area (scale bar, 5 nm). **c** AFM image (inset: height plot). **d** Nitrogen sorption isotherms at 77 K and pore size distribution (inset). **e** Particle size distribution by dynamic light scattering (inset: photograph of colloid solution after 4 months and observed Tyndall effect).

after 3-day Xenon lamp irradiation in water, no obvious changes in PXRD pattern or FT-IR spectrum could be observed for CYANO-COF (Supplementary Fig. 11). These findings revealed the excellent chemical stability and photo-stability of the CYANO-COF.

**Photocatalytic hydrogen evolution reaction.** The UV-vis diffusion reflectance spectroscopy spectrum of CYANO-COF exhibited an absorption band with edges at 627 nm, implying the obvious visible-light responsive nature (Fig. 3a). In comparison with BD-COF, CYANO-COF presented apparent red shifts. It is well established that the absorption edge of π conjugated systems will red shift with the incorporation of chromophore in frameworks[57]. The optical band gaps of CYANO-COF and BD-COF were calculated to be 2.17 and 2.24 eV by Tauc plots, respectively (Supplementary Fig. 12). This result indicated that cyano groups could narrow the band gap, thus increasing the light trapping ability. Furthermore, the conduction band (CB) and valence band (VB) positions of the two COFs estimated by electrochemical Mott–Schottky plots and their optical band gaps were enough for both proton reduction and water oxidation reaction (Supplementary Fig. 12 and Fig. 3b). The positive slope of Mott–Schottky plots indicates typical n-type semiconductor feature for both COFs. CYANO-COF exhibited more negative CB position than that of BD-COF, implying stronger driving force for proton reduction.

We subsequently evaluated the activity of CYANO-COF for HER under visible light (λ > 420 nm). The loading amount of Pt and different types of sacrificial reagent (sodium ascorbate, $Na_2SO_3$ or TEOA) were screened first (Supplementary Table 4 and Supplementary Fig. 13). 1 wt% Pt with 0.1 M ascorbic acid as a sacrificial reagent was the optimized reaction conditions for CYANO-COF. The possible reasons for the high HER activity with ascorbic acid as a sacrificial reagent may be related to the

efficient hole trapping ability of ascorbic acid[58]. Notably, no obvious change in Pt size was observed with the Pt loading used for the test (Supplementary Fig. 13). The volcano curve of HER rate and Pt loading implied the combined effect of the electron trapping for proton reduction and light absorbance by Pt. Under optimized conditions, the average $H_2$ evolution rate of CYANO-COF and BD-COF was 1217 and 39.5 μmol h$^{-1}$ (Fig. 3c), respectively. A 30-fold increase of the $H_2$ evolution rate of CYANO-COF in comparison with BD-COF demonstrated the promotion effect of cyano groups in photocatalytic HER.

Amazingly, the photocatalytic HER rate of CYANO-CON was as high as 2684 μmol h$^{-1}$, more than twice that of CYANO-COF. The BD-CON also showed an increased hydrogen production rate (159 μmol h$^{-1}$), which was four times higher than that of pristine bulk COF. The enhanced hydrogen production rate of the several-layered nanosheet as compared with the bulk COFs was related to the short migration distance of photogenerated charge carriers and also more exposing reaction surface[48,49]. Deuterium isotope experiments were carried out using $D_2O$, and the evaluated gases were detected by mass spectrometry. Nearly all mass-to-charge contributions are $D_2$, indicating that the produced $H_2$ was indeed from water molecules (Supplementary Fig. 14). A high AQE of 82.6% at 450 nm was achieved for CYANO-CON (Fig. 3d). The AQE decreased as the wavelength of the irradiation light increasing, identical to the light absorption properties of CYANO-CON. Interestingly, even when CYANO-CON was irradiated with 650 nm red light, it could still achieve a high AQE of 4.2%, demonstrating its high efficiency in photocatalytic HER. A blank experiment was also performed with only $H_2PtCl_4$ or PVP-protected Pt nanoparticles but no $H_2$ could be detected. This observation proved that Pt served as a co-catalyst, which could not only trap the photogenerated electrons from the semiconductor due to its high work functions but also reduce the activation energy of proton reduction to promote the surface reactions (Supplementary Fig. 15).

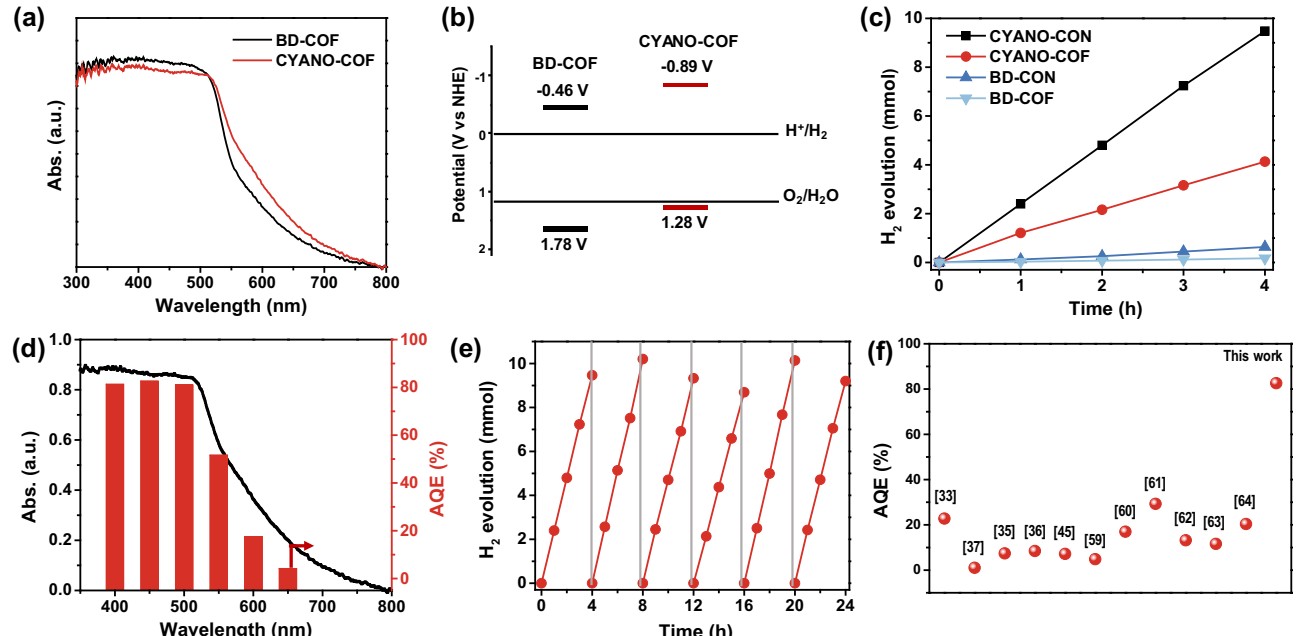

**Fig. 3 Absorption spectra, band positions and photocatalytic H₂ evolution. a** UV-vis DRS spectra of BD-COF and CYANO-COF. **b** Schematic energy band structures of BD-COF and CYANO-COF. **c** Time course of photocatalytic H₂ production for different COFs and CONs (20 mg catalyst in 100 mL water, 1 wt% Pt, 10 mmol ascorbic acid, λ > 420 nm). **d** Wavelength-dependent AQE of photocatalytic H₂ production for CYANO-CON. **e** Cycling stability for CYANO-CON in photocatalytic H₂ production. **f** AQE of HER for state-of-the-art representative COFs and conjugated polymers.

We also compared CYANO-CON to other COFs and polymers reported in the literature for H₂ evolution reaction (Fig. 3f and Supplementary Tables 6 and 7)[33–38,45,59–64]. Clearly, in terms of AQE, CYANO-CON outperforms these reported COF/polymer-based photocatalysts, with the highest AQE value, which is comparable to that of a previously reported star inorganic Pt-PdS/CdS photocatalyst[4]. Moreover, even when comparing H₂ evolution rates normalized by the mass, CYANO-CON is superior to most of the COFs/polymer-based photocatalysts. It is also worth noting that CYANO-CON showed a remarkable photocatalytic activity (83.1 μmol h⁻¹, λ > 420 nm) and an AQE as high as 2.72% at 450 nm even in the absence of Pt co-catalyst (Supplementary Fig. 15 and Supplementary Table 5). Nevertheless, the H₂ production activity of BD-CON was very low (0.63 μmol h⁻¹) without Pt, further confirming the promotion effect of the cyano groups (Supplementary Fig. 15).

The long-term recycling experiment with CYANO-CON as model catalyst showed no obvious decline in H₂ production rate for more than 24 h (Fig. 3e) and ~90% CYANO-CON was recovered after six cycles. The TEM image, PXRD pattern and FT-IR spectrum of CYANO-CON remained almost the same before and after photocatalysis (Supplementary Figs. 16 and 17), signifying the excellent stability of CYANO-CON for photocatalysis. We also drop-casted Pt-CYANO-CON colloid solution onto a glass support (size of 1 cm × 6 cm, Supplementary Fig. 18). Hydrogen bubbles could be clearly observed over CYANO-CON film under visible light irradiation for 10 h (Supplementary Fig. 18 and Supplementary Movie 1). The average HER rate of CYANO-CON film could reach 292 mmol m⁻², much higher than that of a reported COF film[45].

Furthermore, the photocatalytic oxygen evolution reaction (OER) was investigated in this work. Intriguingly, CYANO-CON could catalyze photocatalytic OER to afford an OER rate of 1.933 μmol h⁻¹ with CoₓOᵧ as co-catalyst and AgNO₃ as electron sacrificial reagent (Supplementary Fig. 19). The ¹⁸O-labeled water experiment confirmed that the oxygen was sourced from water.

A control experiment to test the OER activity of La₂O₃ and cobalt nitrate without CYANO-CON was also performed. The results showed that La₂O₃ and cobalt nitrate do not have photocatalytic OER activity under visible light (λ > 420 nm). This low OER rate may be attributed to the less positive VB position of CYANO-CON (1.28 eV vs water oxidizing potentials 1.23 eV, pH = 0) and the sluggish four-electron transfer kinetic process of oxygen generation. It should be noted that although BD-CON has a more positive band position compared to CYANO-CON, the tested OER activity of BD-CON (0.05 μmol h⁻¹) was much lower than that of CYANO-CON (Supplementary Fig. 19). In the TEM images of the used CYANO-CON and BD-CON, the existence of CON nanosheet could still be clearly observed together with irregularly shaped nanoparticles assigned to the photo-deposition of metallic silver nanoparticles, indicating the high stability of CYANO-CON and BD-CON under photocatalytic OER conditions (Supplementary Fig. 17).

**Mechanistic studies to uncover the role of CYANO.** CYANO-CON afforded much higher activity than BD-CON despite the fact that the two CONs have similar BET surface area, pore size and topology structure. The above electrochemical and optical characterization results revealed that the existence of CYANO would lead to an increase of light trapping and a negative shift of CB position. However, these thermodynamic properties did not provide any insights into the exciton and charge carriers properties, which are very important for understanding the photocatalysis process.

To understand the exciton properties of CONs, temperature-dependent PL measurements were carried out to determine the exciton binding energies. The integrated PL intensity of both CONs decreased with increasing temperature from 77 to 253 K, which can be mainly attributed to the thermally activated nonradiative recombination process (Fig. 4a, c)[32,52,65]. Furthermore, based on a simple model, the temperature-dependent PL

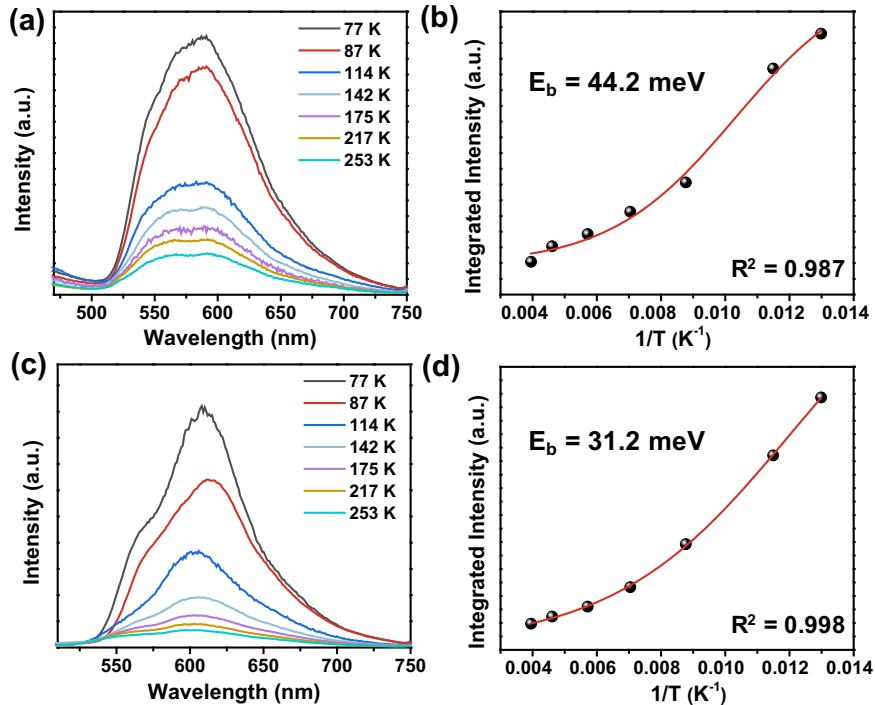

**Fig. 4 Exciton binding energies measurement.** Temperature-dependent PL spectra with excitation wavelength at 380 nm and extracted exciton binding energies of **a**, **b** BD-CON and **c**, **d** CYANO-CON.

intensity of the two CONs can be expressed by the following equation:

$$I(T) = \frac{I_0}{1 + A e^{-E_b/k_B T}}$$

where $I_0$ is the intensity at 0 K, $E_b$ is the binding energy, $A$ is a proportional constant and $k_B$ is the Boltzmann constant[65,66]. By fitting the experimental data, the exciton binding energies of CYANO-CON and BD-CON were estimated to be 31.2 and 44.2 meV, respectively (Fig. 4b, d), demonstrating that the excitons of CYANO-CON were more prone to dissociation than those of BD-CON, and thus improved the ratios of free charge carriers for CYANO-CON and contributed to its high photo-catalytic activity.

In addition, fs-TA measurements were conducted to investigate the difference in photocatalysis between the two COFs[52]. First, a typical colloidal CYANO-CON sample was excited using a 400 nm pump pulse, and the TA spectrum was acquired with a broadband probe pulse (Fig. 5a). The spectrum exhibited a broad negative bleaching signal in the range of 420–540 nm assigned to the ground state bleach (GSB), which indeed coincided to the steady-state absorption spectrum (Supplementary Fig. 20). In addition, the spectrum showed a weak and broad positive signal from 550 to 750 nm (Fig. 5a). This positive signal could be attributed to trapped carriers (so-called "polarons" in polymers), because its formation was complementary to the decay of the GSB signal within 0.8 ps (Fig. 5b). The trapped carriers could be further assigned specifically as trapped holes on the basis of their rapid decay in the presence of the hole scavenger AA. As shown in Supplementary Fig. 21, these trapped holes were transferred to AA in <2 ps, and CYANO-CON exhibited a more efficient hole to AA transfer process than BD-CON. Following the initial hole trapping process, the holes and electrons recombined slowly, leading to the simultaneous decay of the hole signal and a negative broad feature signal centered at ~560 nm. The latter could be assigned to the stimulated emission of the trapped

exciton, on account of its spectral feature that coincided with the steady-state PL spectra (Supplementary Fig. 22).

The lifetime of charge carriers was studied by TA kinetics of CONs. As shown in Fig. 5c, the decay curves for the trapped hole of the two CONs revealed quite different lifetimes. The lifetime of CYANO-CON (14.2 ± 2.3 ps) was three times longer than that of BD-CON (4.3 ± 0.6 ps). Conventionally, a longer charge carrier lifetime decreases the probability of electron-hole recombination which is a competitive and detrimental process in real photo-catalysis system, explaining the high activity of CYANO-CON both in OER and HER. Notably, after quenching the holes with AA, a new broad negative signal ranging from 500 to 750 nm clearly emerged within the ns time scale and could be attributed to the generation of long-lived free electrons (Supplementary Fig. 23). The reduced charge transfer resistance of CONs characterized by electrochemical impedance spectra can also confirm the existence of free electrons under illumination (Supplementary Fig. 24). Fitting kinetics revealed that long-lived electrons of CYANO-CON had a much slower rate constant ($k_e = 1.3$ ns$^{-1}$) than that of BD-CON ($k_e = 5.3$ ns$^{-1}$) (Fig. 5d), which correlated well with H$_2$ production activity. All these observations indicated that the existence of CYANO can effectively extend the lifetime of charge carriers and eventually increase the photocatalytic hydrogen evolution activity.

## Discussion

In summary, we synthesized a CYANO-COF with β-ketene-cyano as D–A pair for photocatalytic HER. An observed thirty-fold increase of H$_2$ evolution rate of CYANO-COF in comparison with BD-COF demonstrated the promotion effect of cyano groups in photocatalytic HER. A more than two-fold increase in photocatalytic HER rate was also observed by decreasing the COF layer number to ~12–15 layers. Interestingly, CYANO-CON achieved an AQE of up to 82.6% at 450 nm, superior to all currently reported polymer semiconductors in photocatalytic HER to our knowledge. CYANO-CON possesses an intrinsically lower

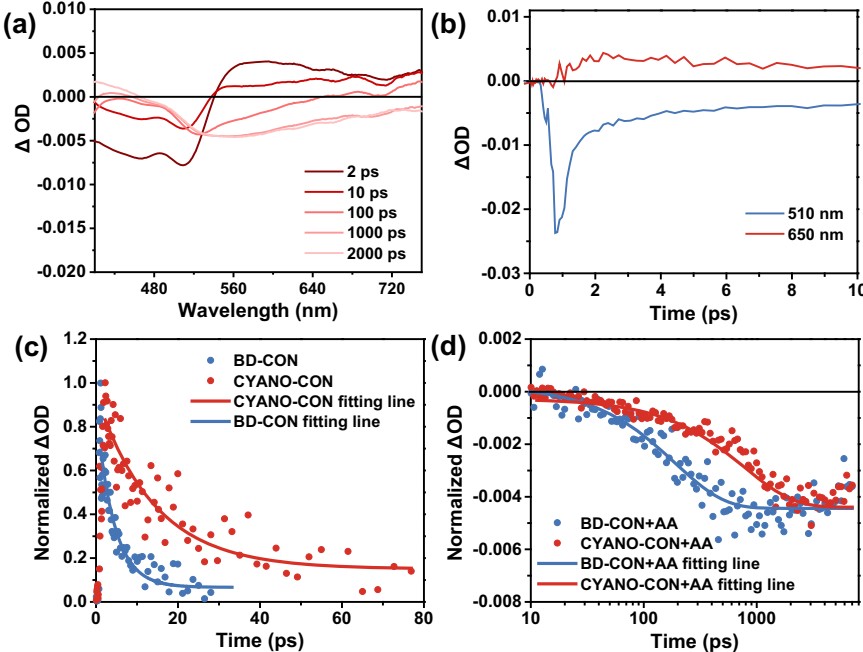

**Fig. 5 Femtosecond transient absorption measurements.** TA details of CYANO-CON and BD-CON pumped at 400 nm. **a** Time slices of the TA spectra of CYANO-CON in water. **b** TA kinetics of CYANO-CON probed at 510 nm (GSB) and 650 nm (trapped hole). Comparison of the kinetics for CYANO-CON and BD-CON **c** probed at 650 nm (0–80 ps for trapped hole) and **d** probed at 650 nm (10–8000 ps for long-lived free electron) in the presence of 0.1 M ascorbic acid.

exciton binding energy in comparison with BD-CON as demonstrated by temperature-dependent PL spectroscopy. Moreover, the fs-TA spectroscopy characterizations revealed that the existence of the cyano group can dramatically extend the lifetime of charge carriers. The high charge separation efficiency could be the main reason for the excellent photocatalytic HER activity of CYANO-CON. This study clearly demonstrates the importance of a suitable D–A pair in the charge separation and migration step of photocatalysis, which will shed light on the development of promising polymer semiconductors.

## Methods

**Synthesis of BD-COF and CYANO-COF.** A 10 mL high-pressure flask was charged with 1,3,5-triformylphloroglucinol (168 mg, 0.8 mmol) and diamine (1.2 mmol). A mixture of 1,4-dioxane and mestitylene (1:1 v/v, 8.0 mL) was added, and the resulting suspension was sonicated at room temperature for 10 min. A 6 M aqueous acetic acid solution (2 mL) was added and the resulting suspension was further sonicated for 10 min. The flask was degassed through three freeze-pump-thaw cycles to remove any dissolved oxygen. After warming it at room temperature, the flask was charged with $N_2$, sealed under positive $N_2$ pressure, and it was then placed into a 120 °C pre-heated oil bath for 3 days. After that, the obtained solid was washed with DMF, acetone and THF for several times and extracted by Soxhlet extraction using THF for 1 day. Finally, the solid materials were dried under vacuum at 120 °C overnight to get the COFs in ~80% isolated yield.

**Synthesis of covalent organic nanosheets (CONs) from COFs by ball milling method.** In all, 150 mg of as-synthesized COFs were placed into a ball mill jar equipped with a suitable amount of zirconia balls (24 balls for Φ = 3 mm and 6 balls for Φ = 6 mm) and the resulting mixture was milled at 50 Hz (planetary ball mill) for 30 min. The obtained CONs were collected without any additional treatment, and the isolated yield of CONs was more than 99%. For TEM and AFM imaging, 1 mg of CONs was dispersed in 10 mL of ethanol, sonicated for 30 min, and subsequently dropped on the carbon-coated copper grid (TEM) and mica (AFM). The solution was allowed to naturally volatilize overnight at room temperature before the measurements.

**Photocatalytic hydrogen evolution.** A flask with a quartz filter was charged with the COFs (20 mg), 0.1 M ascorbic acid water solution (100 mL), and an appropriate amount of $H_2PtCl_6$ as a co-catalyst. The resulting mixture was sonicated for 10 min before degassing by Ar bubbling for 30 min. The reaction system was irradiated with a 300 W Xe lamp (PLS-FX300, Perfectlight) for a specific time using cut-on

filters (λ > 420 nm). Gas samples were taken with a gas-tight syringe (Hamilton 1700) and run on a gas chromatograph (Agilent 8860) equipped with Molecular Sieve 5A column connected to thermal conductivity detector. The generated hydrogen was detected referencing against standard gas with a certain molar amount of hydrogen. Hydrogen dissolved in the reaction mixture was not measured and the slight pressure increase generated by the evolved hydrogen was neglected. After photocatalysis experiments, the photocatalysts were recovered by washing with water and ethanol and then dried under vacuum at room temperature for 5 h for the next round of photocatalysis.

## Data availability

All data generated in this study are provided in the Supplementary Information/Source Data file. Source Data are provided with this paper.

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

## Acknowledgements

This work was supported by the National Natural Science Foundation of China (21733009, 22002162, 21973091) and the Strategic Priority Research Program of the Chinese Academy of Sciences (XDB17020200). J.W. thanks the financial support from the Youth Innovation Promotion Association CAS (2021185). We acknowledge Dr Ting Yang for assistance with the PL measurement. We also gratefully acknowledge Alexis Munyentwali for a careful proofreading of the manuscript. C.L. thanks Dr Fangfang Xu and Xiaomin Ren for helpful discussions.

## Author contributions

C.L. did most of the experiments, structure simulation and wrote the manuscript; J.L. helped with SEM characterization; K.W. helped with TA characterization and analysis; Q.Y., H.L. and J.W. designed and supervised the project. The manuscript was written through the contributions of all authors.

## Competing interests

The authors declare no competing interests.
