## [Peer Review File · Nature Communications]

Covalent organic frameworks with high quantum efficiency in sacrificial photocatalytic hydrogen evolutionREVIEWER COMMENTS

Reviewer #1 (Remarks to the Author):

The manuscript: "Covalent organic frameworks with high quantum efficiency in photocatalytic hydrogen evolution: mediating charge separation", by Li et al, presents photocatalysis in the presence of Pt co-catalysts. The importance of the Pt is however not emphasized sufficiently which creates the impression that all catalysis happens in the COF. Using PL and TA the authors present some convincing evidence for the underlying photophysics. However, the manuscript could improve on this, in particular discussion on whether the Pt accepts/donates electrons to the COF.

Inspecting the TA data in plot 5c and d: I remain skeptical regarding the assignment of free electrons in the spectrum. These assignments in TA are always a bit challenging as there is no clear feature to be expected and the probe is in the optical range in which the interaction with free electrons is less pronounced. The authors should present additional proof of free electrons, for example measuring conductivity/photo-conductivity. Either DC conductivity under CW illumination or ultrafast conductivity, for example optical pump THz probe, would provide conclusive evidence of free charges in the system.

The authors also should explain the purpose of Pt in the catalytical reaction in more detail. I suspect that Pt is the actual catalyst in the reaction? If so that should be clearly emphasized as the main body of the paper discusses COF/CON and not all actually catalytically active species.

A few experimental details are missing in my opinion. In line 140 the authors mention "high-energy ball milling". What was the energy? What are the expected temperature changes? Was any of these properties measured? In the methods section the ball-milling is followed by ultrasound treatment. Sonication is commonly used to exfoliate 2D materials and might also be important here. Please discuss whether the ultrasound or the ball milling is the crucial step for exfoliation and present evidence either way.

A minor point is that it took me a while to understand that CN is just carbon and nitrogen, in the forest of abbreviation this tree is quite hidden, in my opinion. I therefore recommend to once write these elements out. There are also a few other abbreviations I am still not certain what they mean as they are not defined in the manuscript, for example CTFs (line 63). This manuscript is submitted to a broad audience journal and as such I strongly encourage the authors to double check that all abbreviations are explained/needed.

Overall, the English of the manuscript is great; However, there are a few minor typos and I suggest that the authors give it one more proofread. The typos that I noticed, without looking for any, are for example, misspelling of Schottky's name (line 173), or misspelling average in line 216.

Reviewer #2 (Remarks to the Author):

The report by Li et al. presents two materials for sacrificial hydrogen production from water, with one material also studied for oxygen production. Personally, I think that the report of a material that performs both half-reactions is probably the most interesting aspect as there are now many reports of hydrogen production and oxygen evolution seems to be the biggest hurdle for the field to progress. Nevertheless, the hydrogen production efficiency is impressive with high quantum yields over a wide range. The material remains active as a thin-film and the photophysical studies add depth to this study. Overall, it appears clear that the difference between the materials originates from differences in charge-separation efficiencies, which has been shown in through the TA studies.

I have no doubts that this should be published in Nature Communications after a minor revision based

on the comments that I have below:

- 1) The title should include the word 'sacrificial' as this is not water splitting.
- 2) State-of-the-art presentation misses in particular conjugated polymers, which have improved significantly recently.
- 3) There are other examples of cyano-benzene materials that have been reported for hydrogen production.
- 4) Given that particle size appears to play a major role I would expect that this is much more discussed in the introduction. This has been shown by others with conjugated polymers very convincingly.
- 5) The following sentence is missing a verb: 'Still, these COFs do not possess high AQE or photocatalytic HER activity, which is possibly related with the inefficient D-A pair.' There are other sentences that do not make sense to me (e.g. 'Thus, a ketene-CN D-A pair was successfully incorporated in CN-COF, which was further confirmed by the computational study of charge distribution (Fig. 1b).') The whole manuscript requires a bit more attention.
- 6) How do the BET surface area and pore sizes compare to predictions given that an AA stacking is assumed?
- 7) Given the large size of the particles I am not sure if DLS is the best way to analyse the material size. Static light scattering experiments allow for a wider range to be studied.
- 8) Comparing absolute rates as in Fig 3d makes no sense given that all materials were measured on different set-ups with different light sources. As such, I would expect that only quantum efficiencies are compared as these remove the uncertainty of light source intensity. The authors even mention issues surrounding mass normalization in the text, but then discuss this in broad strokes anyways. The whole section from line 198 to 209 needs to be reworked to state explicitly comparisons to the state-of-the-art.
- 9) The band-structure diagram should be presented in the main-text.
- 10) The other material should be tested for oxygen evolution. The reviewer would also expect that the band positions play an important role and the other material would be expected to perform better.
- 11) I wonder about the procedure used for photocatalytic oxygen production experiments. Using a syringe is unsuitable as atmospheric will contaminate the injector and syringe needle. Is this a mistake? Otherwise, these experiments need to be rerun on a suitable system.
- 12) The conclusion that charge separation is important has been shown by others (Cooper, McCulloch + Durrant) using TA. The work should be cited here as similar conclusions are drawn for unbranched conjugated polymer photocatalysts.
- 13) TA studies of the oxygen evolution half-reaction would be extremely interesting too.
- 14) Yields should be stated along the recovered mass of the product.

Reviewer #3 (Remarks to the Author):

In this work, the authors presented the high AQE for photocatalytic H₂ generation using a modified COF. This work is interesting and has done quite a detailed study. This manuscript can be considered after revision. The detailed comments are given below.

- (1) A scale bar is missing in Fig. 2b.
- (2) L133-134: The authors wrote "The hexagonal straight pore feature of CN-CON can be clearly observed (Fig. 2a)" - No hexagonal pore is observed in Fig. 2a.
- (3) The authors have conducted photocatalytic H₂ evolution in presence of cocatalyst (Pt) to achieve high AQE. Therefore it is also necessary to estimate the AQE without cocatalyst (Pt).
- (4) For OER, the authors have used 100 mg La₂O₃ as a buffering agent and cobalt nitrate as cocatalyst, both are considered good OER catalysts. What are the views of the author about this? OER performance of CN-CON is shown here but not the other. As the VB edge of BD-COF is more positive it

might be possible that it gives higher OER performance than CN-CON. Thus experiments must be conducted without a cocatalyst to know the precise contribution of the synthesized materials.

(5) Ascorbic acid, as the sacrificial reagent, shows dramatic increase in H₂ evolution performance as compared to other sacrificial reagents. What is/are the possible reason for such high activity with ascorbic acid?

(6) TEM study of the catalyst after HER and OER is preferred.

Reviewer #1 (Remarks to the Author):

The manuscript: “Covalent organic frameworks with high quantum efficiency in photocatalytic hydrogen evolution: mediating charge separation”, by Li et al, presents photocatalysis in the presence of Pt co-catalysts. The importance of the Pt is however not emphasized sufficiently which creates the impression that all catalysis happens in the COF. Using PL and TA the authors present some convincing evidence for the underlying photophysics. However, the manuscript could improve on this, in particular discussion on whether the Pt accepts/donates electrons to the COF.

Response: Thanks for your suggestion. The title of the manuscript was changed to “Covalent organic frameworks with high quantum efficiency in sacrificial photocatalytic hydrogen evolution: mediating charge separation” in the revised manuscript and the discussion of the role of Pt in the photocatalytic H₂ production was also added in the revised manuscript (details see the response to Question 2).

Question 1. Inspecting the TA data in plot 5c and d. I remain skeptical regarding the assignment of free electrons in the spectrum. These assignments in TA are always a bit challenging as there is no clear feature to be expected and the probe is in the optical range in which the interaction with free electrons is less pronounced. The authors should present additional proof of free electrons, for example measuring conductivity/photo-conductivity. Either DC conductivity under CW illumination or ultrafast conductivity, for example optical pump THz probe, would provide conclusive evidence of free charges in the system.

Response: Thanks for your suggestion. We agree with the Reviewer that the assignment free electron signal is not easy due to the complicated π - π conjugated structure and charge separation process of conjugated polymers. In this work, the hole signal is clearly assigned by the comparison of the spectra with and without sacrificial reagent (Supplementary Figure 21). The quenched hole cannot recombine with free electron to emit fluorescence and interfere with the assignment of free electron signal. Therefore, in the case of excluding other possibilities, we consider that the new negative signal in Supplementary Figure 21 is probably to be a free electron signal.

We totally agree with the Reviewer that the ultrafast spectroscopy (optical pump THz probe) can indeed provide critical evidence for free electrons. However, currently, our institution does not have this equipment to allow us to complete this characterization. Due to the pandemic of Covid-19, it is also not easy for us to seek help from another related research groups in other place.

Remedially, we carried out the AC impedance test of COFs under illumination and dark conditions (Supplementary Figure 24). The results showed that the charge transfer impedance of COFs can be significantly reduced under illumination condition, which more or less proves the existence of free electrons. We added this in the revised

manuscript.

Supplementary Figure 21. TA kinetics of (a) BD-CON and (b) CYANO-CON probed at 650 nm (trapped hole). In the presence of 0.1 M AA, the hole signal disappeared with the appearance of a negative absorption peak assigned to an ultra-fast component stemming from hole transfer from VB of CONs to the hole scavenger.

Supplementary Figure 24. (a) Electrochemical impedance spectra (EIS) of BD-CON and CYANO-CON were carried out under dark and illumination (> 420 nm, 15 A Xe lamp), with an AC potential frequency ranging from 0.1 Hz to 100 kHz. In the equivalent circuit (inset), R_s represents the circuit series-resistance, CPE1 is the capacitance phase element of the semiconductor-electrolyte interface, and R_{ct} is the charge transfer resistance across the interface, (b) Simulated R_s and R_{ct} values of CONs for electrochemical impedance test.

Question 2. The authors also should explain the purpose of Pt in the catalytical reaction in more detail. I suspect that Pt is the actual catalyst in the reaction? If so that should be clearly emphasized as the main body of the paper discusses COF/CON and not all actually catalytically active species.

Response: Thanks for your suggestion. In photocatalytic hydrogen evolution reaction (HER), noble metal or inexpensive metal are generally regarded as “co-catalysts”, and

the corresponding views have been widely reported (Acc. Chem. Res. 2013, 46, 8, 1900-1909; Chem. Soc. Rev., 2014, 43, 7787-7812). In other words, Pt NPs can promote activity of the photocatalyst, but it does not have photocatalytic activity by itself.

In order to avoid misunderstanding, we carried out a series of blank and control experiments (Supplementary Figure 15). The following discussions “Very interestingly, CYANO-CON shows obvious photocatalytic activity ($83.1 \mu\text{mol h}^{-1}$) even in the absence of Pt co-catalyst and the AQE is as high as 2.72% at 450 nm without Pt (Supplementary Table 5). But the H_2 production activity of BD-CON is very low ($0.63 \mu\text{mol h}^{-1}$) without Pt, further confirming the promotion effect of cyano groups (Supplementary Fig. 15). The blank experiment was also performed with only H_2PtCl_4 or PVP-protected Pt nanoparticles and no H_2 could be detected, proving that Pt acts as co-catalyst, which could not only trap the photo-generated electrons from the semiconductor due to its high work functions but also reduce the activation energy of proton reduction to promote the surface reactions (Supplementary Fig. 15).” were added in the revised manuscript.

Supplementary Figure 15. Time course of photocatalytic H_2 production for blank control experiment with (a) 0.2 mg Pt/PVP or H_2PtCl_4 , (b) 20 mg CONs without cocatalyst (100 mL water, 10 mmol ascorbic acid, $\lambda > 420 \text{ nm}$).

Question 3. A few experimental details are missing in my opinion. In line 140 the authors mention “high-energy ball milling”. What was the energy? What are the expected temperature changes? Was any of these properties measured? In the methods section the ball-milling is followed by ultrasound treatment. Sonication is commonly used to exfoliate 2D materials and might also be important here. Please discuss whether the ultrasound or the ball milling is the crucial step for exfoliation and present evidence either way.

Response: Thank you for such a sincere suggestion. We performed the ball milling with QM-3SP2 and the ball milling conditions are similar to the literatures (Nat. Mater. 2013, **12**, 1130-1136; J. Am. Chem. Soc. 2017, **139**, 5842–5848; ACS Nano 2019, **13**, 5893–5899). Unfortunately, we cannot provide the information of the energy and the expected temperature changes during the ball milling process. We can

expect that the elevated temperature was not high enough to destroy the chemical composition of COFs, due to the almost same FT-IR spectra of CONs and COFs. We changed the “High-energy ball milling” to “ball milling” in the revised manuscript.

The control trials were performed to exfoliate the COFs with sonication treatment for different times. The results show that morphology of CYANO-COF remained unchanged with different sonication time (Supplementary Figure 7). Therefore, the ball milling is the crucial step for the exfoliation of CYANO-COF. We added this discussion in the revised manuscript.

Supplementary Figure 7. SEM images of (a) CYANO-COF and (b) BD-COF after sonication for 30 min. SEM image of (c) CYANO-COF after sonication for 24 h. (scale bar: 1 μm)

Question 4. A minor point is that it took me a while to understand that CN is just carbon and nitrogen, in the forest of abbreviation this tree is quite hidden, in my opinion. I therefore recommend to once write these elements out. There are also a few other abbreviations I am still not certain what they mean as they are not defined in the manuscript, for example CTFs (line 63). This manuscript is submitted to a broad audience journal and as such I strongly encourage the authors to double check that all abbreviations are explained/needed.

Overall, the English of the manuscript is great; However, there are a few minor typos and I suggest that the authors give it one more proofread. The typos that I noticed, without looking for any, are for example, misspelling of Schottky’s name (line 173), or misspelling average in line 216.

Response: Thanks for your suggestion. We checked all abbreviations and added missing explanations. To avoid the misunderstanding, the CN-COF and CN-CON were respectively changed to CYANO-COF and CYANO-CON in the revised manuscript. We have polished the English in the revised manuscript.

Reviewer #2 (Remarks to the Author):

The report by Li et al. presents two materials for sacrificial hydrogen production from water, with one material also studied for oxygen production. Personally, I think that the report of a material that performs both half-reactions is probably the most

interesting aspect as there are now many reports of hydrogen production and oxygen evolution seems to be the biggest hurdle for the field to progress.

Nevertheless, the hydrogen production efficiency is impressive with high quantum yields over a wide range. The material remains active as a thin-film and the photophysical studies add depth to this study. Overall, it appears clear that the difference between the materials originates from differences in charge-separation efficiencies, which has been shown in through the TA studies.

I have no doubts that this should be published in Nature Communications after a minor revision based on the comments that I have below:

Response: We appreciate the reviewer very much for the positive and valuable comments.

Question 1. The title should include the word ‘sacrificial’ as this is not water splitting.

Response: Thanks for your suggestion. We changed the title to “Covalent organic frameworks with high quantum efficiency in sacrificial photocatalytic hydrogen evolution: mediating charge separation” in the revised manuscript.

Question 2. State-of-the-art presentation misses in particular conjugated polymers, which have improved significantly recently.

Response: Thanks for your suggestion. The references about progress of conjugated polymers were cited as refs 40, 41, and 44 in the introduction of the revised manuscript.

Question 3. There are other examples of cyano-benzene materials that have been reported for hydrogen production.

Response: Thanks for your suggestion. We cited some other cyano-benzene materials for hydrogen production in introduction as ref. 52 in the revised manuscript.

Question 4. Given that particle size appears to play a major role I would expect that this is much more discussed in the introduction. This has been shown by others with conjugated polymers very convincingly.

Response: Thanks for your suggestion. We added the recent advances in the particle size effect in introduction and corresponding references are cited as refs 42, 43 in the revised manuscript.

Question 5. The following sentence is missing a verb: ‘Still, these COFs do not possess high AQE or photocatalytic HER activity, which is possibly related with the inefficient D-A pair.’ There are other sentences that do not make sense to me (e.g.

‘Thus, a ketene-CN D-A pair was successfully incorporated in CYANO-COF, which was further confirmed by the computational study of charge distribution (Fig. 1b).’) The whole manuscript requires a bit more attention.

Response: Thanks for your suggestion. We have polished the English in the revised manuscript.

Question 6. How do the BET surface area and pore sizes compare to predictions given that an AA stacking is assumed?

Response: Thanks for your suggestion. The Connolly surface area of AA stacking model was calculated to be $2667 \text{ m}^2 \text{ g}^{-1}$ by using Atom Volumes & Surfaces tools in Materials Studio software. The predicted pore size of CYANO-COF was estimated to be about 2.1 nm by using the Measure/Change Distance tools in Materials Studio software. We added the following discussion “The Brunauer–Emmett–Teller (BET) surface area of CYANO-COF measured by nitrogen sorption isotherms at 77 K was $559 \text{ m}^2 \text{ g}^{-1}$, lower than the theoretical BET surface area ($2667 \text{ m}^2 \text{ g}^{-1}$) with AA stacking model. The different experiment and theoretical value is generally observed for COFs possibly due to the structure distortion. The pore size of CYANO-COF is distributed from 1.0 to 2.5 nm calculated by nonlocal density functional theory (NLDFT) method (Supplementary Table 3, Supplementary Fig. 5), which agreed well with predicted AA stacking pore size (Supplementary Fig. 6).” in the revised manuscript.

Supplementary Figure 6. (a) Top view of calculated Connolly surface and predicted pore size of CYANO-COF AA stacking model, (b) comparison of Connolly surface area and experimental BET surface area.

Question 7. Given the large size of the particles I am not sure if DLS is the best way to analyse the material size. Static light scattering experiments allow for a wider range to be studied.

Response: Thanks for your suggestion. Static light scattering (SLS) is a common characterization to determine the size of particle suspensions in the sub- μm and supra- μm ranges, via the Lorenz-Mie and Fraunhofer diffraction formalisms,

respectively. However, both DLS and SLS can only give the statistical particle size distribution (equivalent average size), and the real shape and size details of irregular nanosheets cannot be recognized more comprehensively. We further measured the particle size of the CYANO-CON by AFM characterizations in a large scale (30 μm). As shown in Supplementary Figure 10, CYANO-CON presented irregular nanosheet topography. By measuring 200 particles (excluding a few elongated nanosheets), the particle size of the nanosheet varied in the range of 200 nm to 3.5 μm , and the most probable particle distribution was similar to DLS results. We added this result in the revised manuscript.

Supplementary Figure 10. (a) AFM image and (b) particle size distribution (measuring 200 particles) of CYANO-CON.

Question 8. Comparing absolute rates as in Fig 3d makes no sense given that all materials were measured on different set-ups with different light sources. As such, I would expect that only quantum efficiencies are compared as these remove the uncertainty of light source intensity. The authors even mention issues surrounding mass normalization in the text, but then discuss this in broad strokes anyways. The whole section from line 198 to 209 needs to be reworked to state explicitly comparisons to the state-of-the-art.

Response: Thanks for your suggestion. We reproduced the figure by comparison of only quantum efficiencies. The whole section from line 198 to 209 has been rewritten in the revised manuscript.

Question 9. The band-structure diagram should be presented in the main-text.

Response: Thanks for your suggestion. The band-structure diagram has been added as Fig 3b in the revised manuscript.

Question 10. The other material should be tested for oxygen evolution. The reviewer would also expect that the band positions play an important role and the other material would be expected to perform better.

Response: Thanks for your suggestion. We checked the photocatalytic OER activity

of BD-CON. The result showed that BD-CON gave much lower oxygen evolution activity than CYANO-CON (Supplementary Figure 19). In addition to the band position, the charge separation efficiency and hole lifetime of CONs also greatly influence the photocatalytic activity. This discussion was added in the revised manuscript.

Supplementary Figure 19. (a) Time course of photocatalytic O₂ production for CYANO-CON and BD-CON with 1wt % Co(NO₃)₃ as co-catalyst (20 mg catalyst in 100 mL water, 100 mg La₂O₃, 0.5 mmol AgNO₃, λ > 420 nm). (b) Co content dependent oxygen evolution activity of CYANO-CON. (c) Isotope labeling experiment was conducted by using H₂¹⁸O instead of H₂O for photocatalytic oxygen evolution which exhibits the evolution of ¹⁸O₂ gas.

Question 11. I wonder about the procedure used for photocatalytic oxygen production experiments. Using a syringe is unsuitable as atmospheric will contaminate the injector and syringe needle. Is this a mistake? Otherwise, these experiments need to be rerun on a suitable system.

Response: Thanks for your reminding. We tested the photocatalytic oxygen production with standard vacuum system again as shown in supporting information (Supplementary Figure 25e). The standard vacuum equipment showed similar OER activity with the syringe needle one, because we calibrated the standard curve of air and deducted this disturbing value in original experiment. In the revised manuscript, all the OER evaluation data were collected by using the standard vacuum system.

Supplementary Figure 25e. The system for photocatalytic oxygen production.

Question 12. The conclusion that charge separation is important has been shown by others (Cooper, McCulloch + Durrant) using TA. The work should be cited here as similar conclusions are drawn for unbranched conjugated polymer photocatalysts.

Response: Thanks for your suggestion. We cited the above work as ref. 44 in the revised manuscript.

Question 13. TA studies of the oxygen evolution half-reaction would be extremely interesting too.

Response: Thanks for your suggestion. We are also very curious about TA studies of the oxygen evolution half-reaction. However, the addition of La_2O_3 and AgNO_3 makes the whole system unable to maintain a stable, clear and bright solution state. Reliable transient absorption signals cannot be obtained in such a complex suspension system.

Question 14. Yields should be stated along the recovered mass of the product.

Response: Thanks for your suggestion. Yield of recovered CYANO-CON was ~90%, and this discussion was added in the revised manuscript.

Reviewer #3 (Remarks to the Author):

In this work, the authors presented the high AQE for photocatalytic H_2 generation using a modified COF. This work is interesting and has done quite a detailed study. This manuscript can be considered after revision. The detailed comments are given below.

Question 1. A scale bar is missing in Fig. 2b.

Response: Thanks for your suggestion. The scale bar was added in Fig. 2b in the revised manuscript.

Question 2. L133-134: The authors wrote "The hexagonal straight pore feature of CYANO-CON can be clearly observed (Fig. 2a)" - No hexagonal pore is observed in Fig. 2a.

Response: Thanks for your reminding. This sentence was rewritten as "The ordered arrangement of mesopores could be clearly observed in the HRTEM image of CYANO-CON (Fig. 2a)".

Question 3. The authors have conducted photocatalytic H_2 evolution in presence of cocatalyst (Pt) to achieve high AQE. Therefore it is also necessary to estimate the AQE without cocatalyst (Pt).

Response: Thanks for your suggestion. We added the photocatalytic activity and AQE of CYANO-CON without Pt in Supplementary Table 5 and Supplementary Figure 15. in the revised manuscript. Very encouragingly, CYANO-CON showed the AQE of 2.61 % at 400 nm in the absence of Pt, further confirming the potential application of CYANO-COF in photocatalysis.

Supplementary Figure 15. Time course of photocatalytic H₂ production for blank control experiment with (a) 0.2 mg Pt/PVP or H₂PtCl₄, (b) 20 mg CONs without cocatalyst (100 mL water, 10 mmol ascorbic acid, $\lambda > 420$ nm).

Supplementary Table 5. Wavelength-dependent AQE of photocatalytic H₂ production for CYANO-CON without co-catalyst.

λ (nm)	400	450	500	550	600	650
H ₂ evolution rate ($\mu\text{ mol h}^{-1}$)	8.98	17.4	21.3	20.1	5.75	0.99
Number of photons ($\times 10^{20} \text{ h}^{-1}$)	4.14	7.71	9.58	13.7	13.3	11.6
AQE (%)	2.61	2.72	2.67	1.77	0.52	0.10

Question 4. For OER, the authors have used 100 mg La₂O₃ as a buffering agent and cobalt nitrate as cocatalyst, both are considered good OER catalysts. What are the views of the author about this? OER performance of CYANO-CON is shown here but not the other. As the VB edge of BD-COF is more positive it might be possible that it gives higher OER performance than CYANO-CON. Thus, experiments must be conducted without a cocatalyst to know the precise contribution of the synthesized materials.

Response: We totally agree with you that La₂O₃ may also show photocatalytic OER

activity. We performed a control experiment to test the OER activity of La_2O_3 and cobalt nitrate without the CYANO-CON. The results show that La_2O_3 and cobalt nitrate do not show photocatalytic OER activity under visible light ($> 420 \text{ nm}$). We also tested the OER activity of BD-CON. The BD-CON showed much lower oxygen evolution activity than CYANO-CON (Supplementary Figure 19a and Supplementary Table 8). According to your suggestion, the photocatalytic OER activity of CYANO-CON and BD-CON was also tested without co-catalyst. Unfortunately, both CONs are inactive for OER without cocatalyst. We added the above discussions in the revised manuscript.

Supplementary Figure 19. (a) Time course of photocatalytic O₂ production for CYANO-CON and BD-CON with 1wt % $\text{Co}(\text{NO}_3)_3$ as co-catalyst (20 mg catalyst in 100 mL water, 100 mg La_2O_3 , 0.5 mmol AgNO_3 , $\lambda > 420 \text{ nm}$). (b) Co content dependent oxygen evolution activity of CYANO-CON. (c) Isotope labeling experiment was conducted by using H_2^{18}O instead of H_2O for photocatalytic oxygen evolution which exhibits the evolution of $^{18}\text{O}_2$ gas.

Supplementary Table 8. Oxygen evolution performance of COF by using different additives.^[a]

COFs	Additives		O ₂ evolution rate (μmol/h)
	La_2O_3	$\text{Co}(\text{NO}_3)_2$	
-	+	+	0
CYANO-CON	-	-	0
CYANO-CON	+	-	0
CYANO-CON	+	+	1.9
BD-CON	+	+	0.05
BD-CON	+	-	0

^[a] Reaction conditions: 20 mg of COFs was suspended in 100 mL of an aqueous solution with different additive (100 mg La_2O_3 , 1wt % $\text{Co}(\text{NO}_3)_2$) and 0.5 mmol AgNO_3 as sacrificial agent, irradiated by a 300 W Xe lamp ($\lambda > 420 \text{ nm}$).

Question 5. Ascorbic acid, as the sacrificial reagent, shows dramatic increase in H₂ evolution performance as compared to other sacrificial reagents. What is/are the possible reason for such a high activity with ascorbic acid?

Response: The possible reasons for the high HER activity with ascorbic acid may be related with the fast hole trapping ability of ascorbic acid. We added this discussion in the revised manuscript.

Question 6. TEM study of the catalyst after HER and OER is preferred.

Response: Thanks for your suggestion. The TEM images of photocatalyst after HER and OER were added in Supplementary Figure 17. No obvious changes in the TEM images of CYANO-CON and BD-CON were observed after HER reaction. In the TEM images of the used CYANO-CON and BD-CON, the existence of CON nanosheet could still be clearly observed together with irregularly shaped nanoparticles assigned to the in situ formed OER co-catalyst, indicating the high stability of CYANO-CON and BD-CON under photocatalytic OER conditions. We added the above discussions in the revised manuscript.

Supplementary Figure 17. The TEM images of (a,b) CYANO-CON and (c,d) BD-CON after (a,c) HER and (b,d) OER.

REVIEWERS' COMMENTS

Reviewer #1 (Remarks to the Author):

The authors addressed my concerns and revised the manuscript accordingly. I think that the manuscript improved and can therefore recommend publication in its current form.

Reviewer #2 (Remarks to the Author):

The revision has addressed my comments well and I only have a few additional comments:

- 1) I am still somewhat confused what pressures were used for which experiments. It would perhaps help to include these explicitly in the experimental section and be mentioned in the main-text too.
- 2) The TEMs post-photocatalytic oxygen evolution reaction are surprising as typically the deposition of metallic silver is observed. Why is this not the case here?
- 3) The experimental procedures need to specific stating yields, amounts obtained, and amounts/molarities used in the experiment (e.g. photocatalytic experiments section).

Reviewer #3 (Remarks to the Author):

The authors have substantially improved the manuscript as per the suggestions/comments of the reviewers. Thus the revised manuscript may be considered for publication.

Reviewer #1 (Remarks to the Author):

The authors addressed my concerns and revised the manuscript accordingly. I think that the manuscript improved and can therefore recommend publication in its current form.

Response: We thank the reviewer for this positive comment and support the publication of our manuscript.

Reviewer #2 (Remarks to the Author):

The revision has addressed my comments well and I only have a few additional comments.

Response: We appreciate the reviewer very much for the positive and valuable comments for the improvement of our manuscript.

Question 1. I am still somewhat confused what pressures were used for which experiments. It would perhaps help to include these explicitly in the experimental section and be mentioned in the main-text too.

Response: Thanks for your suggestion. The photocatalytic hydrogen evolution was conducted under positive pressure. The detailed information for the pressure used for HER was added in the experimental section as “The generated hydrogen was detected referencing against standard gas with a certain molar amount of hydrogen. Hydrogen dissolved in the reaction mixture was not measured and the slight pressure increase generated by the evolved hydrogen was neglected.” The photocatalytic oxygen evolution was conducted under reduced pressure, and the procedure for photocatalytic oxygen evolution has been added in the revised supplementary information as follows: “The mixture was sonicated for 10 min and the solution was evacuated several times to completely remove air. The reaction was then illuminated with a 300 W Xe light source for the time specified using cut-on filters ($\lambda > 420$ nm) under reduced pressure.”

Question 2. The TEMs post-photocatalytic oxygen evolution reaction are surprising as typically the deposition of metallic silver is observed. Why is this not the case here?

Response: Thanks for your suggestion. The elemental mappings for CYANO-CON and BD-CON after OER were provided as Supplementary Figure 17. From the TEM pictures and corresponding elemental mappings, the deposition of metallic silver could be clearly observed.

Supplementary Figure 17. (a,d) TEM images of (a) CYANO-CON and (d) BD-CON after HER, (b,e) TEM images of (b) CYANO-CON and (e) BD-CON after OER. (c,f) Elemental mapping of (c) CYANO-CON and (f) BD-CON after OER.

Question 3. The experimental procedures need to specific stating yields, amounts obtained, and amounts/molarities used in the experiment (e.g. photocatalytic experiments section).

Response: Thanks for your suggestion. The detailed information for the experiments was added in the revised manuscript.

Reviewer #3 (Remarks to the Author):

The authors have substantially improved the manuscript as per the suggestions/comments of the reviewers. Thus the revised manuscript may be considered for publication.

Response: We thank the reviewer for this positive comment and support the publication of our manuscript.